# Pregestational Prediabetes Induces Maternal Hypothalamic–Pituitary–Adrenal (HPA) Axis Dysregulation and Results in Adverse Foetal Outcomes

**DOI:** 10.3390/ijms25105431

**Published:** 2024-05-16

**Authors:** Mathuli Ngema, Nombuso D. Xulu, Phikelelani S. Ngubane, Andile Khathi

**Affiliations:** School of Laboratory Medicine & Medical Sciences, University of KwaZulu-Natal, Westville, Private Bag X54001, Durban 4041, KwaZulu Natal, South Africa; 218022309@stu.ukzn.ac.za (M.N.); 215019278@stu.ukzn.ac.za (N.D.X.); ngubanep1@ukzn.ac.za (P.S.N.)

**Keywords:** maternal hypothalamic–pituitary–adrenal (HPA) axis, foetal HPA axis, pregestational prediabetes, pups

## Abstract

Maternal type 2 diabetes mellitus (T2DM) has been shown to result in foetal programming of the hypothalamic–pituitary–adrenal (HPA) axis, leading to adverse foetal outcomes. T2DM is preceded by prediabetes and shares similar pathophysiological complications. However, no studies have investigated the effects of maternal prediabetes on foetal HPA axis function and postnatal offspring development. Hence, this study investigated the effects of pregestational prediabetes on maternal HPA axis function and postnatal offspring development. Pre-diabetic (PD) and non-pre-diabetic (NPD) female Sprague Dawley rats were mated with non-prediabetic males. After gestation, male pups born from the PD and NPD groups were collected. Markers of HPA axis function, adrenocorticotropin hormone (ACTH) and corticosterone, were measured in all dams and pups. Glucose tolerance, insulin and gene expressions of mineralocorticoid (MR) and glucocorticoid (GR) receptors were further measured in all pups at birth and their developmental milestones. The results demonstrated increased basal concentrations of ACTH and corticosterone in the dams from the PD group by comparison to NPD. Furthermore, the results show an increase basal ACTH and corticosterone concentrations, disturbed MR and GR gene expression, glucose intolerance and insulin resistance assessed via the Homeostasis Model Assessment (HOMA) indices in the pups born from the PD group compared to NPD group at all developmental milestones. These observations reveal that pregestational prediabetes is associated with maternal dysregulation of the HPA axis, impacting offspring HPA axis development along with impaired glucose handling.

## 1. Introduction

Foetal programming is a process whereby a stimulus or insult at a critical period in development results in permanent adaptation of the organism’s structure or physiology [1,2,3]. Evidence in foetal programming studies has shown that foetal overexposure to endogenous glucocorticoids may underpin the link between early life events and later disease [4,5,6,7]. It is proposed that dysregulation of the hypothalamic–pituitary–adrenal (HPA) axis determines foetal exposure to stress hormones, influencing foetal development and programming the foetal HPA axis [8,9,10,11,12].

In humans, the physiologically active glucocorticoid (GC) is cortisol, whereas in rodents, it is corticosterone [13,14]. During pregnancy, the maternal HPA axis experiences significant changes, with the placenta secreting corticotropin-releasing hormone (CRH), which further elevates adrenocorticotropin hormone (ACTH) and cortisol levels [15,16,17]. This creates a positive feedback loop where maternal cortisol stimulates placental CRH synthesis, ultimately resulting in higher glucocorticoid levels [4,18]. In addition, previous research has reported that despite the increasing circulating levels of glucocorticoid, the diurnal secretion of corticosterone is maintained throughout pregnancy [19,20,21].

Some cases of maternal adversity, such as maternal stress, have been associated with prolonged activation and dysregulation of the maternal HPA axis, leading to elevated plasma ACTH and cortisol levels [9,22,23,24]. Additionally, research has indicated that T2DM shares similarities with maternal stress conditions during pregnancy, including the persistent activation and dysregulated function of the HPA axis with elevated glucocorticoid levels [25,26,27,28,29,30]. Rapid economic development, urbanisation, sedentary lifestyles, and a Westernised diet have led to a rising burden of 463 million (20–79 years) adults living with T2DM, resulting in an increasing proportion of pregnancies complicated by diabetes [31,32,33,34]. Moreover, research has shown that diabetic pregnant women have elevated levels of cortisol alongside exacerbated dysfunction of the hypothalamic–pituitary–adrenal (HPA) axis when compared to non-diabetic pregnant women [35]. Given that T2DM is a complicated and multifaceted disease caused by a mix of genetic and environmental risk factors, it is considered a stressor to the human body [36,37,38,39,40]. Studies show that excessive levels of maternal can overwhelm the enzymatic barriers that effectively prevent excessive foetal exposure to maternal GCs, therefore exposing foetuses to excess glucocorticoids [5,29,30,41]. Studies have shown that pregnancies affected by T2DM in conjunction with uncontrolled hyperglycaemia, increased oxidative stress, hypertension, vascular diseases, and increased glucocorticoids, among other complications, exhibit intrauterine growth restriction (IUGR), often manifested as a low birth weight [42,43]. Studies have also shown that excessive foetal exposure to GCs is associated with the downregulation of foetal GR and MR and impairment of the feedback regulation of the HPA axis in both infancy and adulthood [5,13,44,45]. Cross-sectional research has indicated a connection between low birth weight and disrupted functioning of the HPA axis, leading to elevated levels of GC in adulthood [10,46,47,48]. In addition, the association between a low birth weight and the development of T2DM was first reported in studies by Hales et al., who demonstrated a several-fold increase in the incidence of glucose intolerance and T2DM in adult men who were born with low birth weight compared with those born with normal birth weight [49,50]. Additionally, research suggests that individuals born with a low birth weight are often associated with catch-up growth with an increased risk for various non-communicable diseases (NCDs) such as hypertension, cardiovascular diseases, and mental disorders in adulthood, aligning with the developmental origins of health and disease (DOHaD) hypothesis [48,51,52,53,54,55,56].

Furthermore, due to the transgenerational risk and foetal complications associated with diabetic pregnancies and the rising frequency of diabetes in young adults [57,58,59,60,61,62,63,64], researchers have redirected their focus towards the concept that intrauterine periods may be one of the contributing factors to an increased risk of metabolic diseases in adults [58,59,60,61,62]. However, studies have shown that T2DM is often preceded by an early-onset condition known as prediabetes [65,66,67]. Prediabetes is a condition in which blood glucose concentrations are higher than normal but do not meet the diagnostic criteria for T2DM [68,69,70]. Studies show that prediabetes is predicted to affect 453.8 million people by 2030 and usually 5–10% progress to T2DM each year due to asymptomatic characteristics [71,72]. A diet-induced animal model for prediabetes was established in our laboratory and it was found to mimic the human condition [73,74,75]. In addition, this animal model showed similarities in pathophysiology with T2DM, including dysregulation in the HPA axis associated with increased basal corticosterone and impaired regulation of their GR and MR in male animals [76]. This raised the question of whether the difficulties associated with maternal stress and preexisting T2DM pregnancies and foetal programming of foetal HPA axis are also present during prediabetes and whether basal corticosterone and ACTH levels in prediabetic dams may impact foetal HPA axis development. Therefore, using this animal model, the study sought to investigate the effects of pregestational prediabetes on maternal HPA axis function and its effects on postnatal offspring development.

## 2. Results

### 2.1. Dams’ Oral Glucose Tolerance (OGT) Response and Glycated Haemoglobin (HbA1c) Concentrations

The OGT (Figure 1a) and HbA1c concentrations (Figure 1b) in the non-prediabetic (NPD) female group (n = 6) and prediabetic (PD) female group (n = 6) at 36 weeks are shown in Figure 1. In the OGTT (Figure 1a), the blood glucose concentration is significantly higher in the PD group at time 0 when compared to the NPD group. The blood glucose concentration in the PD group remained significantly higher when compared to the NPD group throughout the 2 h test. The HbA1c concentration (Figure 1b) was significantly higher in the PD group when compared to the NPD group.

### 2.2. Dams’ Hypothalamic–Pituitary–Adrenal (HPA) Axis Components

The dams’ ACTH (Figure 2a) and corticosterone (Figure 2b) concentrations in the non-prediabetic (NPD) female group (n = 6) and prediabetic (PD) female group (n = 6) at 21 days postpartum. The ACTH concentration (Figure 2a) was significantly higher in the PD group when compared to the NPD group. Similarly, the corticosterone concentration (Figure 2b) was significantly higher in the PD group when compared to the NPD group.

### 2.3. The Dams’ Homeostasis Model Assessment (HOMA)-IR, HOMA-S, and HOMA-β Indices, and Plasma Adrenocorticotrophic Hormone (ACTH) and Corticosterone Concentrations

The dams’ glucose handling was measured by assessing insulin sensitivity and beta-cell function through the insulin resistance (HOMA-IR), insulin sensitivity (HOMA-S%), and Beta-cell secretory capacity (HOMA-β%) indices, along with plasma ACTH and corticosterone concentrations. Table 1 shows the HOMA-IR index assessment for glucose handling in the non-pre-diabetic (NPD) and pre-diabetic (PD) female groups (n = 6 per group) with measurements of plasma ACTH and corticosterone concentrations. The HOMA-IR value for NPD was within the insulin-sensitive range (<1.0), while the PD group had a significantly higher HOMA-IR value compared to the PD, which was in the range of significant insulin resistance. HOMA-S percentage was significantly lower in the PD group in comparison to the NPD group, while the HOMA-β percentage of the PD group was significantly higher compared to the NPD group. The results further showed that there was an increase in plasma ACTH and corticosterone concentrations in the PD group compared to the NPD.

### 2.4. Pups Hypothalamic–Pituitary–Adrenal (HPA) Axis Components

The ACTH (Figure 3a) and corticosterone (Figure 3b) concentrations of the pups born from the non-prediabetic (NPD) pregnant female group (n = 6 per week) and pups born from the prediabetic (PD) pregnant female group (n = 6 per week) at weeks 3, 6, and 16 of the experimental periods. The ACTH concentration (Figure 3a) was significantly higher in the PD groups when compared to the NPD groups throughout the experimental weeks. Similarly, the corticosterone concentration (Figure 3b) was significantly higher in the PD groups when compared to the NPD groups throughout the experimental weeks.

### 2.5. Pups Hippocampal Glucocorticoid Receptors (GR) and Mineralocorticoid Receptor (MR)

The hippocampal GR (Figure 4a) and MR (Figure 4b) gene expressions were measured in non-stressed pups born from the non-prediabetic (NPD) pregnant female group (n = 6 per week) and prediabetic (PD) pregnant female group (n = 6 per week) at weeks 3, 6, and 16 of the experimental periods. The PD groups had a half-fold decrease in GR (Figure 4a) gene expression relative to the NPD groups in all experimental weeks. The PD groups had a two-fold increase in MR (Figure 4b) gene expression relative to the NPD groups in all experimental weeks.

### 2.6. Pups Adrenal Gland Weight

The adrenal gland weight of the pups born from the non-prediabetic (NPD) pregnant female group (n = 6 per week) and pups born from the prediabetic pregnant (PD) female group (n = 6 per week) at weeks 3, 6, and 16 of the experimental periods. The adrenal weight in the PD groups was significantly higher when compared to the NPD groups throughout the experimental weeks (Figure 5).

### 2.7. Pups Bodyweights

The body weights of the pups born from the non-prediabetic (NPD) pregnant female group (n = 6) and pups born from the prediabetic (PD) pregnant female group (n = 6) on day 7 (Figure 6a), and at weeks 3, 6, and 16 (Figure 6b) of the experimental periods. On day 7 (Figure 6a), the body weight of the PD group was significantly lower when compared to the NPD group. At week 3 (Figure 6b), body weight in the PD group showed no significant difference when compared to the NPD group. At week 6 (Figure 6b), the body weight of the PD group was significantly higher when compared to the NPD group. Lastly, at week 16 (Figure 6b) the body weight in the PD group showed no significant when compared to the NPD group.

### 2.8. Pups Oral Glucose Tolerance (OGT) Response

The OGT of pups born from non-prediabetic (NPD) pregnant female group (n = 6 per week) and prediabetic (PD) pregnant female group (n = 6 per week) at weeks 3, 6, and 16 of the experimental periods. The OGTT blood glucose concentration was significantly higher in the PD groups at time 0 when compared to the NPD groups in all experimental weeks. The blood glucose concentration in the PD groups remained significantly higher throughout the 2-hr test when compared to the NPD groups in all experimental weeks (Figure 7).

### 2.9. The Pups’ HOMA-IR, HOMA-S, and HOMA-β Indices, and Plasma Adrenocorticotrophic Hormone (ACTH) and Corticosterone Concentrations

The pup’s glucose handling was measured by assessing insulin sensitivity and beta-cell function through the HOMA-IR, HOMA-S, and HOMA-β indices, along with plasma ACTH and corticosterone concentration at weeks 3, 6, and 16. Table 2 shows the HOMA-IR index assessment for glucose handling in the pups born from non-pre-diabetic (NPD) and pre-diabetic (PD) female groups (n = 6 per group) with measurements of plasma ACTH and corticosterone concentration. HOMA-IR value for NPD was within the insulin-sensitive range (<1.0) except at week 16, while the pups born from the PD group had a significantly higher HOMA-IR value, which was in the range of significant insulin resistance. The HOMA-S percentage was significantly lower in the pups of the PD group in comparison to the pups of the NPD group while the HOMA-β percentage in the pups born from the PD group was significantly higher compared to the pups born from the NPD group except for week 16. The results further showed that there was an increase in plasma ACTH and corticosterone concentrations in the pups born from the PD group compared to the pups born from the NPD group.

## 3. Discussion

Foetal programming, a response to adverse foetal conditions, leads to lasting adaptations altering organ growth, physiology, and metabolism, thus increasing adult disease risk [77,78]. Excessive GC exposure in utero, often due to maternal HPA axis dysregulation, has been shown to link early events with later diseases such as hypertension, cardiovascular diseases, T2DM, and mental disorders [5,79,80]. During normal pregnancy, the maternal HPA axis undergoes significant changes, yet diurnal GC secretion remains maintained [9,81]. There are studies suggesting that T2DM exhibits resemblances to maternal stress conditions during pregnancy, such as dysregulated HPA axis with increased levels of GC [25,26,27,28]. Foetal exposure to excess maternal GCs causes growth restriction and programmed life-long changes in HPA axis activity, which increases the risk of developing T2DM and cardiometabolic diseases in adult life [45,81,82]. Several studies have suggested that the onset of complications associated with T2DM begins during the prediabetic state [65,66,67]. An experiment in our lab established a diet-induced prediabetic animal model and showed similarities with humans, including dysregulation in the function of the HPA axis [76]. However, no studies have yet shown the influence of pre-existing prediabetes during pregnancy on the maternal–foetal HPA axis interaction. Therefore, this study aimed to investigate the effects of pregestational prediabetes on maternal HPA axis function and its effects on postnatal offspring development.

In non-diabetic individuals, glucose homeostasis is tightly regulated with fasting plasma glucose (FPG) maintained at 3·9–5·6 mmol/L and postprandial glucose level of less than 7.8 mmol/L [83]. In the postprandial state, elevated blood glucose concentration stimulates pancreatic beta β cells to produce adequate insulin enough to clear glucose from the bloodstream through the insulin signalling pathway [84,85]. Studies show that during T2DM, the insulin signalling pathway is disturbed primarily due to chronic hyperglycaemia, which then contributes to pancreatic β-cell dysfunction and insulin resistance [86,87]. Studies show that T2DM is also associated with IGF and IGT and elevated HbA1c ≥6.5% [88]. Studies show that prediabetes can be diagnosed by at least two of these characteristics: impaired fasting glucose (IFG) (5.5–6.9 mmol/L), impaired glucose tolerance (IGT) (7.8–11.0 mmol/L) and elevated glycated haemoglobin A1c (HbA1c) (5.7–6.4%) [89,90]. In the present study, there was a significant increase in the fasting plasma glucose concentration before glucose loading and a failure of blood glucose concentration post-glucose load to return to baseline following a 2 h OGT test in the PD group, suggesting the presence of IGF and IGT in Figure 1a. In the present study, the PD group had a significant increase in HbA1c concentration when compared to the NPD group in Figure 1b. The results align with previous research indicating that elevated plasma glucose concentrations, also seen in PD and T2DM, result in non-enzymatic glycation of haemoglobin [73,91,92]. This glycation process occurs throughout the entire 120-day lifespan of red blood cells through an Amadori reaction, forming a stable and irreversible ketoamine linkage [73]. These findings in our results indicate that the levels of glucose in the blood and the length of time that red blood cells are exposed to glucose are responsible for the production of HbAc1. This suggests that glucose utilisation in insulin-dependent peripheral tissues such as skeletal muscles is decreased, suggesting there is some insulin resistance in the tissues [93,94]. In this study, insulin sensitivity and beta cell function were assessed using the Homeostasis Model assessment (HOMA), showing insulin resistance (HOMA-IR), reduced insulin sensitivity (HOMA-S), and increased beta cell function (HOMA of β-cell function) in the PD group in Table 1. Studies show that increased pancreatic β-cell function is associated with the production and increased secretion of insulin as a compensatory mechanism to a high blood glucose concentration circulating in the peripheral tissues [95]. The coexistence of IGF and IGT, elevated HbAc1 levels, insulin resistance, reduced insulin sensitivity, and increased beta cell function in our PD group indicated the induction of prediabetes at 36 weeks. A prior investigation conducted in our lab revealed that male animals developed prediabetes after 20 weeks of being fed a diet high in fat and carbohydrates [73]. Our current study extends this timeframe by an additional 16 weeks. A previous study has attributed this to several physiological disparities, including genetic and hormonal differences such as progesterone and oestrogen, that have been shown to exert protective effects that may have delayed the induction of prediabetes in females [96].

During pregnancy, the regulation of the maternal HPA axis undergoes dramatic changes, such as regulating stress response and maintaining homeostasis for both the mother and the developing foetus [22,24]. The HPA axis controls the diurnal secretion of GCs, which play a crucial role in foetal development [9,19]. Physiological active glucocorticoid is known as cortisol in humans and corticosterone in rats [13,14]. Studies show that elevated cortisol levels during pregnancy enhance the inhibition of insulin activity, leading to decreased insulin sensitivity and increased insulin resistance in skeletal muscle by reducing glucose utilisation [97,98]. Studies show that patients and animals with poorly controlled or uncontrolled diabetes commonly have both elevated basal ACTH and high levels of GCs due to the altered regulation of the HPA axis [99,100]. In addition, according to a previous study, diabetic pregnant women exhibit higher cortisol concentration with exacerbated HPA axis dysfunction compared to non-diabetic pregnant women [35]. Studies show that T2DM in pregnancy shows similar findings seen among maternal obesity, depressed and stressed pregnant women with prolonged activation and dysregulated function of the HPA axis with elevated glucocorticoid levels [101,102]. Studies show that pregnancy in women with T2DM worsens especially in those who already have other complications such as uncontrolled hyperglycaemia, hypertension, or vascular diseases [103,104]. In the present study, we evaluated HPA axis activity by measuring two components of the HPA axis under basal non-stressful conditions and found that both plasma ACTH levels and corticosterone concentrations were significantly increased in the PD pregnant female group when compared to the NPD pregnant female group in Figure 2. These results corroborated the previous study that found a dysregulation of the HPA axis in male prediabetic animals evidenced by the elevated basal corticosterone concentration along with the unchanged ACTH concentration in non-stressed conditions [76]. In the rats, late pregnancy and the postpartum period have been associated with reduced basal activity of the HPA axis with decreased ACTH and corticosterone [105,106,107]. Therefore, the basal increased concentrations of ACTH and corticosterone may be an indication that, indeed, the pre-existing prediabetic state in pregnancy maintained the impaired negative feedback and HPA axis dysregulation.

During non-diabetic pregnancy, the placenta CRH, which initiates a complex feed-forward loop [108,109,110]. Placental CRH activates the maternal HPA axis, resulting in the production of cortisol. At the same time, maternal cortisol increases the synthesis of placental CRH, leading to an increase in cortisol levels, which is a vicious cycle [15,16,17]. In addition, during development, foetuses need glucocorticoids for various aspects of brain development and at late gestational lung maturation [111,112]. However, approximately 5–10% of cortisol must only pass the placenta to the foetus, and this is due to the partial placental protective barrier 11β-Hydroxysteroid Dehydrogenase 2 (11β-HSD 2) [113]. Studies in diabetic pregnancy, similarly to maternal obesity, depressed, and stressed pregnancy, report that high maternal cortisol is associated with the dysregulation of placental CRH, which results in a vicious cycle of high maternal cortisol [102,114,115]. Studies show that high maternal cortisol in pregnancy can cross the placenta by overwhelming the placental barrier into the foetal compartment which may directly affect foetal development [116,117,118]. Furthermore, studies show that exposure to adverse maternal cues, such as high glucocorticoids during critical developmental periods, has been shown to increase the risk of altered HPA axis function, stress-related conditions such as depression, cardiometabolic diseases and T2DM later in life [119,120,121,122]. The above is in line with the foetal programming hypothesis. The exposure to glucocorticoid in utero is thought to compromise foetal brain development, specifically the prefrontal cortex, hippocampus, and amygdala, brain areas associated with regulating the HPA axis [80,123,124]. In addition, research indicates that excessive cortisol exposure during early gestation triggers an early shift from tissue accretion to differentiation, thereby reducing foetal growth in various vital organs such as the brain, heart, liver, kidney, and adrenal glands [125,126,127]. This process often leads to the clinical manifestation of intrauterine growth restriction (IUGR), characterised by the development of growth-retarded foetuses [128,129]. The diagnosis of IUGR is assigned to infants with a birth weight below the 10th percentile for gestational age [130,131]. In diabetic pregnancy, IUGR is observed most in patients with vasculopathy (retinal, renal, or chronic hypertension) [42,50,132,133]. The association between a low birth weight and elevated plasma cortisol concentrations, hypertension, cardiovascular diseases, T2DM, and mental disorders has been documented by epidemiological studies [134,135]. However, the impact of maternal dysregulated HPA axis function in pregestational prediabetes and its influence on foetal HPA axis and postnatal offspring development has not yet been explored. Therefore, this leads to the next phase of the study.

The foetal pituitary matures first, with foetal HPA activity beginning at mid-gestation [19,136]. The actions of the foetal HPA axis are essential in foetal development, maturation, and homeostasis and eventually prepare for the survival of the neonate [136,137]. After birth, the HPA axis is able to regulate responses to adverse conditions, acting on the metabolism of carbohydrates, proteins and lipids, and participating in anti-inflammatory effects and suppression of the immune response [138,139]. A study found that maternal diabetes with high cortisol levels is linked to changes in the development of the brain’s cortical, neuroendocrine system by reducing the number of hippocampal neurons [140]. This is accompanied by disruptions in the foetal HPA axis, resulting in structural changes such as cortical thinning, enlarged amygdala, and impaired development of the foetal cerebellum [141,142]. The modifications in brain structures caused by cortisol have been consistently linked to impairments in both cognitive and emotional development, as well as basal high cortisol levels in children [140,141,142]. However, the exact molecular and cellular mechanism by which diabetes during pregnancy affects the development of the brain is still unknown [143]. However, previous studies conducted in maternal stress pregnancy show that GC exposure alters hippocampal glucocorticoid receptor density and sensitivity, permanently altering the set-point and HPA axis regulation [144,145]. Studies show that these are observed from the neonatal, prepubertal and post-pubertal periods and appear to persist through to adulthood with increased fasting cortisol levels associated with upregulated HPA axis function [146,147,148,149]. Studies show that there is increased adrenocortical function in children in the juvenile period who were exposed to GC in utero with increased fasting cortisol concentrations in adults [150,151]. The upregulation of postnatal HPA function in other species may reflect changes in the HPA axis at the level of the hypothalamus, pituitary or adrenal gland itself [152,153].

Therefore, with the maternal dysregulation in HPA axis function associated with high corticosterone concentration in pregnancy in our study and the associated consequences shown in the literature. In the present study, we evaluated ACTH and corticosterone concentration components of the HPA axis of the pups born from the PD group and found a significant increase in all developmental stages when compared to pups born from the NPD group in Figure 3. These results indeed correlate with literature that offspring born of maternal diabetes and/or maternal stress exhibit increased corticosterone in animals [154,155,156]. However, previous studies have shown that a prolonged and continual increase in glucocorticoids travels to the brain, where constant, elevated corticosterone levels in the highly regulated brain cause constant activation of the HPA axis, thus leading to HPA axis hyperactivity especially in non-stressed conditions [157,158]. Therefore, the prolonged increase in ACTH and corticosterone concentrations in all developmental stages in our study may have resulted in HPA axis hyperactivity, as our animals were not stressed.

In addition, HPA axis activity is modulated by a feedback regulation of glucocorticoids exerted by two different types of receptors, namely the MR and the GR [158,159]. The relationship between MR and GR is also critical to negative feedback as the two receptors act co-ordinately to reduce corticosterone secretion by inhibiting ACTH secretion following exposure to stress and maintaining homeostatic balance at rest [160,161]. This balance is critical for the normal function of the HPA axis [162]. However, excess glucocorticoid exposure to maternal stress has been shown to reduce the number of both glucocorticoid and mineralocorticoid receptors in the hippocampus, subsequently altering the set point of the foetal HPA axis, which is evident after birth [30,123,129]. Hence, the present study evaluated hippocampal GR and MR gene expressions. In this study, the pups born from the PD group had significant decreased GR gene expression, while MR gene expression had significantly increased, consistently in all developmental stages when compared to pups born from the NPD group in Figure 4. The decreased GR expression correlates with the previous study and supports that maternal increased corticosterone in our study may have overwhelmed the placenta barriers and crossed over the placenta and crossed the blood–brain barrier to the foetal brains and occupied GR, leading to GR resistance due to constant HPA axis activity. However, the increase in MR gene expression contrasted with other studies that show a decrease in its expression. Studies show that excess glucocorticoid exposure during utero can lead to changes in DNA methylation patterns, histone modification, and microRNA expression that influence gene expression patterns [44,163]. These epigenetic changes may specifically upregulate MR gene expression and downregulate GR expression as a regulatory response to cope with the persistently elevated levels of glucocorticoid in utero and even after birth [46,164]. Therefore, the imbalances of GR and MR gene expression in our result may have been due to compensatory mechanisms that occurred during utero and persisted after birth to emerging adulthood. Our study supports the hypothesis that high maternal corticosterone or cortisol during pregnancy exposes the foetus to excess glucocorticoids, which promotes persistent changes in the HPA axis function during development, as evidenced by increased ACTH and corticosterone, and imbalances in GR and MR gene expressions. Furthermore, studies show that the adrenal glands may undergo adrenocortical hypertrophy as a result of increased production of corticosterone, as seen in this study [165,166]. This excessive corticosterone secretion has been shown to lead to enlargement of the adrenal glands over time, which correlates with the observed increased adrenal gland weight in pups born from the PD group compared to the NDP group in all the developmental stages in Figure 5.

Moreover, other studies have provided further mechanistic insight into HPA axis programming [167,168,169,170]. A study showed that in the white population, some studies showed maternal cortisol level is a predicted factor for foetal birth weight [171]. A study in diabetic pregnant rats demonstrated a foetal growth retardation/IUGR evident as low birth weight [172]. The findings of this study were also similar to those of depressed and stressed pregnant women. According to previous studies, elevated activity of the HPA axis, specifically characterised by higher levels of ACTH and high levels of glucocorticoids in the blood, is observed in both children and adults who were born with a low birth weight [135,173,174,175]. In addition, previous studies have shown that about 30% of all infants with a low birth weight show catch-up growth during the first 2 years of life, and this is to compensate for their genetically determined growth trajectory [176,177,178]. The detrimental effects of catch-up growth in humans have been associated with the development of glucose intolerance, insulin resistance, T2DM, hypertension, and cardiovascular disease in adulthood [179,180,181]. In the present study, the pups born from the PD group had a significantly lower body weight when compared to pups born from the NPD group Figure 6a. The results corroborated previous findings that showed that high maternal glucocorticoid was also observed in the study, which disturbed foetus development and reduced foetal growth, which is evident in lower body weight. However, in this study, the pups born from the PD group had no significant change in body weight at week 3, known as the juvenile period, corresponding to starting 2 years of human age in rats and at emerging adults’ periods, whereas at the prepubertal period, there was a significant increase in body weight when compared to pups born from NPD group in Figure 6b. Therefore, we deduced that the body weight in the developmental stages showed an absence of catch-up growth due to the body weight discrepancy. Studies show that the absence of catch-up growth may play an important role in protecting the animals from adverse metabolic outcomes in the long term and preventing the catch-up growth deterioration effect [182,183].

Moreover, studies show that elevated maternal GC levels during pregnancy influence both the programming of the foetal HPA axis and metabolic pathways such as glucose metabolism [9,46]. This programming can lead to alterations in the development of tissues such as skeletal muscle involved in glucose homeostasis, as a result of foetal programming during the period of growth restriction also observed in diabetic pregnancies [184,185]. In addition, studies show that offspring born to mothers with diabetes exhibited a greater prevalence of IGT. A study shows that the incidence of IGT increased from 9.4% to 17.4% at juvenile age in children who were born to diabetic mothers [186]. Several studies show that low-birth-weight offspring have been associated with glucose intolerance, insulin resistance, and T2DM in adulthood [55,187]. Hence, in our study, we evaluated glucose tolerance in the pups. In the present study, there was impaired fasting glucose as there was a significant gradual increase in blood glucose concentration in the PD group of all developmental stages in Figure 7. There was also evidence of impaired glucose tolerance in the PD group as the blood glucose concentrations remained higher and a failure of blood glucose concentration to return to baseline after the 2-hour test in all developmental stages. Glucocorticoids have been shown to inhibit pancreatic-β cells from secreting insulin directly, impair insulin-mediated glucose uptake, and interfere in the insulin signalling cascade in peripheral tissues such as skeletal muscle, thus inducing diabetogenic effects and insulin resistance [188,189]. However, studies show that non-diabetic individuals have the ability to counteract the insulin resistance caused by glucocorticoids by either enhancing pancreatic β-cell activity or increasing insulin secretion and sensitivity [190,191]. In the present study, we assessed insulin resistance, pancreatic β-cell and insulin sensitivity, and our results demonstrated insulin resistance, increased pancreatic β-cell, and reduced insulin sensitivity in pups born from the PD group when compared to the NDP group in Table 2. As a result, the observed increased corticosterone, insulin resistance, increased pancreatic β-cell, and reduced insulin sensitivity alongside IGT implies that in the present study, the compensatory mechanism failed to counteract glucocorticoid-induced insulin resistance, resulting in hyperglycaemia. Interestingly, in the NDP group, at week 16, we observed an early-onset insulin resistance with elevated pancreatic β-cell consistent with high corticosterone concentration compared to the other weeks. Studies show that normal glucose-tolerant offspring may have insulin resistance involvement and may be attributable to higher body weight [59]. Therefore, we speculate that the observed onset of insulin resistance in the normal group may be due to the gradual increased body weight and lack of physical activity associated with the increased corticosterone when compared to the other groups. Furthermore, according to previous studies, men born with low birth weight had a several-fold-higher incidence of glucose intolerance, hypertension and type 2 diabetes in adulthood as opposed to those who were born with a normal birth weight [49,50]. In addition, the increased systolic blood pressure seen in supplemental data may predispose our offspring to hypertension correlating with previous research shown in Appendix A. Therefore, the results in this study show that our lower-body-weight offspring may have a greater risk of developing hypertension, T2DM in adulthood and other diseases such as metabolic diseases or psychiatry-related disorders. 

## 4. Materials and Methods

### 4.1. Animals and Housing

All animal experimentation was approved by the Animal Research Ethics Committee (AREC) of the University of Kwa-Zulu Natal (AREC/032/020D). Three-week-old female Sprague Dawley rats were bred and housed in the Biomedical Research Unit (BRU) of the University of Kwa-Zulu Natal and were used in the study. The animals were maintained under standard laboratory conditions of constant temperature (22 ± 2 °C), carbon dioxide (CO_2_) content (<5000 p.m.), relative humidity (55 ± 5%) and illumination (12 h light/dark cycle, lights on at 07h00). The noise level was maintained at less than 65 decibels approved. The animals were allowed access to food and fluids ad libitum. The animals acclimatised to their new environment for 1 week while consuming standard rat chow and tap water before the induction of pre-diabetes by exposure to a well-established experimental diet (HFHC). The experimental design was divided into three distinct phases (A, B, and C), which are outlined in the following paragraphs.

### 4.2. Induction of Prediabetes

In phase A of the study, female Sprague Dawley rats (150–180 g) were randomly assigned to two diet groups, groups A and B (n = 6 per group). Experimental pre-diabetes was induced in female Sprague–Dawley rats using a protocol previously described [73]. To summarise, group A, labelled as the non-prediabetic group (NPD), was given a standard rat chow diet with normal water for a 36-week period. The animals in group B were given a high-fat, high-carbohydrate (HFHC) diet with 15% fructose added for the same duration of 36 weeks. After a duration of 36 weeks, the diagnostic criteria of the American Diabetes Association (ADA) were applied to identify pre-diabetes in all animals. This involved identifying animals exhibiting pre-diabetic indicators, such as fasting blood glucose levels ranging from 5.6 to 7.1 mmol/L, oral glucose tolerance test (OGTT) 2 h glucose levels between 7.1 and 11.1 mmol/L, and glycated haemoglobin (HbA1c) levels between 5.7% and 6.4%, which were used as additional diagnostic criteria for prediabetes. Animals meeting these criteria were categorised as pre-diabetic, while those with measurements below the pre-diabetic thresholds were considered non-pre-diabetic.

### 4.3. Oral Glucose Tolerance Response

An oral glucose tolerance test was performed on all animals to assess their glucose tolerance response. This test was completed after carbohydrate loading, following a well-established laboratory technique [192]. After fasting for 16 h, glucose levels were determined at time 0. Then, a monosaccharide syrup was administered orally using an 18-gauge gavage needle that is 38 mm long and curved, with a 21/4 mm ball end (Able Scientific, Canning Vale, Australia). The glucose concentration was determined by collecting blood using the tail-prick method and measuring glucose concentrations using a OneTouch select glucometer (Lifescan, Mosta, Malta) [193]. Glucose levels were subsequently assessed at 15-, 30-, 60-, and 120-min following carbohydrate loading.

### 4.4. Mating

This led to phase B of the study. Prior to mating, all 12 female Sprague–Dawley rats underwent assessment for the proestrus stage through vaginal smear analysis under a microscope. Those in the proestrus stage were permitted to mate with non-prediabetic male Sprague–Dawley rats. Confirmation of pregnancy and assignment of gestational day (GND) 0 were determined the following morning by the presence of a vaginal plaque or vaginal smear containing spermatozoa observed under a microscope [194]. The male rats were then removed from the cage and male rats were returned to their cage after successful mating [195]. Pregnant rats were then observed until the end of gestation (21 days). This marked the end of phase B of the study.

### 4.5. Male Pups Were Collected for the Study

Phase C of the study began when the pups were born naturally at gestational day 21 and kept with their dams. However, on day 7, the animals were immediately collected, weighed, and immediately returned to the dams. According to previous studies, day 7 represents a neonatal/newborn development period which loosely correlates with similar developmental stages in humans (summarised in Table 3 [196,197,198]. The animals were kept with their mothers for a period of 21 days/3 weeks under standard laboratory conditions (for temperature and humidity) in a 12 h day and 12 h night cycle. This 21-day/3-week cycle allows rats to undergo crucial stages of development, including growth socialisation and the establishment of basic behaviours [199,200,201]. Additionally, the regular day/night cycle helps regulate their circadian rhythms, which are important for overall health and well-being [202,203,204]. After 21 days/3 weeks, the animals were weaned, and 18 male Sprague Dawley pups born from the non-prediabetic (NPD) female group and 18 male pups born from the prediabetic (PD) female group were collected. Afterwards, the dams were subjected to a period of fasting in order to evaluate the oral glucose tolerance (OGT) test and then euthanised for further investigations. This was considered phase one. In the second part of the investigation, the pups were divided into three independent experimental weeks, namely week 3, week 6, and week 16, with six pups in each group assigned to either the NPD or PD group. These animals were given unlimited access to standard chow and normal drinking water throughout their experimental periods. With regard to the number of weeks chosen in our study, studies show that there are six recognised developmental time periods in rats [205,206]. These include the neonatal, infantile, juvenile, peripubertal, late pubertal, and emerging adulthood, which are recognised at weeks 3, 6, and 16 in our study, respectively. These stages have been shown to roughly correspond to similar developmental stages in humans [207,208,209,210,211,212]. Table 3 provides a concise overview of the developmental phases in humans and rats, allowing for a comparison of their respective ages. In addition, at each of the different time points, 6 animals from each group were fasted for 16 h to assess fasting glucose and OGT performance test. Subsequently, the animals were euthanised for terminal investigations at each subsequent week of investigation.

#### Summarised Flow Chart

The methodology flow chart illustrates the phase-to-phase methodology employed in the study shown in Figure 8. Rectangle shapes with A, B, and C represent phases of the study. Phase A shows the induction of prediabetes in female Sprague Dawley rats. In Phase B, the prediabetic rats were mated, became pregnant, and gave birth. In Phase C, the male pups were collected and assigned to 3 developmental milestones (3, 6, and 16 weeks) [213,214,215]. In addition, the animals’ body weights were evaluated and an oral glucose tolerance test (OGTT) was conducted each week of the experiment prior to euthanising the animals for further analysis.

### 4.6. The Homeostasis Model Assessment

At the end of each experimental week (3, 6, and 16), the Homeostasis Model Assessment was utilised to quantify HOMA-IR, HOMA-S, and HOMA-β indices. These indices are used to evaluate insulin resistance, insulin sensitivity, and beta-cell function capability. The calculations were performed using the HOMA2 Calculator v2.2.3 application [216]. Homeostasis model assessment (HOMA) expresses insulin resistance as a HOMA-IR value. A value of less than 1.0 indicates insulin sensitivity, whereas a value more than 1.9 indicates early insulin resistance and a value greater than 2.9 indicates significant insulin resistance. Insulin sensitivity is quantified using the HOMA-S% index, where a larger percentage indicates more insulin sensitivity in the individual. Beta-cell secretory ability is quantified as HOMA-β%, with higher values indicating more insulin secretion by beta-cells to regulate blood glucose levels.

### 4.7. Blood Collection and Tissue Harvesting

For blood collection, all animals were sacrificed using the guillotine including the dams and blood was collected into the separated pre-cooled heparinized tubes and was centrifuged (Eppendorf centrifuge 5403, Hamburg, Germany) at 4 °C 530× *g* for 15 min. The separated plasma was stored at −80 °C in a Bio Ultra freezer (Snijders Scientific, Holland, The Netherlands) for Biochemical analysis. Following blood collection, the hippocampus was removed and placed in pre-cooled Eppendorf containers and snap-frozen in liquid nitrogen before storage in a Bio Ultra freezer (Snijers Scientific, Tilburg, The Netherlands) at −80 °C.

### 4.8. Biochemical Analysis

Plasma insulin, adrenocorticotropic hormone (ACTH) for dams and pups’ concentrations and HbA1c concentrations for dams were measured using their respective rat competitive-ELISA kits (Elabscience Biotechnology Co., Ltd., Wuhan, China) according to the manufacturer’s instructions. The kits included a micro-ELISA plate that was coated with antibodies specific to each of the parameters measured. Standards and samples were pipetted into the appropriate wells of the micro-ELISA plate and incubated for 90 min at 37 °C. This was followed by adding the relevant biotinylated detection antibody (100 μL). After 60 min incubation at 37 °C, avidin–horseradish peroxidase conjugate (100 μL) was added to each microplate well. After a further 30 min incubation at 37 °C, the unbound components were washed away using the wash buffer provided. Substrate solution (100 μL) was added to each microplate well and after 15 min incubation at 37 °C, the stop solution (50 μL) was added. Optical density was measured using a nano spectrophotometer 47 (BMG Labtech, Ortenburg, Germany) at 450 nm. The concentrations of each parameter in the samples were extrapolated from a standard curve.

Corticosterone concentrations for dams and pups were measured in plasma using different Linco-plex kits (Millipore, Darmstadt, Germany). Metabolic hormone kits (Merck-Millipore, Darmstadt, Germany) were used to quantify plasma and Corticosterone concentrations. Assays were performed according to the manufacturer’s instructions. Concentrations were read on Luminex^®^ (50 µL, 50 beads per bead set) using a Luminex^®^ 200™, HTS, FLEXMAP 3D^®^, MAGPIX^®^ instrument (Thermo Fisher Scientific Inc., Waltham, MA, USA) with xPONENT^®^ software (version 3.1.971.0). All samples, quality controls (QCs) and inter-plate control were duplicated. Concentrations of all the analytes in the QCs were within the expected ranges and the inter-plate variation was below 20%. The data generated were managed using Bio-Plex Manager Software, version 4.1.1.

### 4.9. Glucocorticoid Receptor and Mineralocorticoid Receptor Gene Expression via Real-Time-PCR

The hippocampus tissues collected at 3 weeks, 6 weeks, and 16 weeks were homogenised and total RNA was isolated using a ReliaPrep miRNA Cell and Tissue Miniprep System (Promega, Madison, WI, USA). The purity and concentration of RNA was determined by Nanodrop 2000 (Thermo Scientific, Roche, South Africa) was used to determine the purity and concentration of RNA. A purity ratio (A260/A280) of 1.7–2.1 was considered acceptable for conversion of RNA to cDNA following the manufacturer’s instructions (GoTaq^®^ 2-Step RT-Qpcr System as a cDNA synthesis kit, Promega, USA). Total RNA was reverse-transcribed to cDNA. To perform the PCR amplification on ROCHE LightCycler96 (Roche, South Africa), the BIO-RAD iTaq Universal SYBR Green I master mix was used. Primer sequences (Metabion, Germany) used in this study are listed in Table 4 below. The Rattus primer sequences were blasted to verify the primer sequence and their accession number. PCR was performed using the following cycling conditions: an initial denaturation cycle at 95 °C for 10 min followed by PCR, which consisted of 45 cycles at 95 °C for 30 s, 65 °C for 30 s, and a final elongation at 72 °C for 30 s with a single fluorescence measurement. Melting curve analysis was performed at 95 °C for 30 s, 65 °C for 20 s, and 95 °C at a ramp rate of 0.05 °C/s and a continuous fluorescence measurement, followed by a final cooling step at 40 °C for 60 s. The RT-qPCR results were analysed using the 2^−ΔΔCq^ comparative method relative to the control groups. The housekeeping gene used in this study was Glyceraldehyde-3-phosphate dehydrogenase (GAPDH).

### 4.10. Statistical Analysis

All data are expressed as means ± standard error of means (SEM). GraphPad Prism Instant Software (version 8.00, GraphPad Software, San Diego, CA, USA) was used for statistical analysis. All terminal data were analysed using the normality and lognormality test and a student *t*-test to assess differences between control and experimental groups. Values of *p* < 0.05 were considered statistically significantly different between the compared groups.

## 5. Study Limitations and Future Recommendations

One of the limitations of this study is that we were unable to obtain dam placentas for the measurement of placental CRH and partial protective barrier of a glucocorticoid known as 11β-Hydroxysteroid Dehydrogenase 2 (11β-HSD 2) activity. This limitation arose because rats, like many other mammals, typically consume their placenta immediately after giving birth, a behaviour known as placentophagy. Future studies also need to look at the physiological response to stress in pups born from dams with pregestational prediabetes. We also recommend that future studies use hyperinsulinemic clamps for animals for better detection of insulin sensitivity.

## 6. Conclusions

In conclusion, the findings of this study showed that pregestational prediabetes may be associated with maternal dysregulated function of the HPA axis, as evidenced by the elevated ACTH and corticosterone concentrations in the dams. The findings of the study further show that maternal dysregulation in the HPA axis alters the set-point and development of the offspring’s HPA axis, as evidenced by the elevated ACTH and corticosterone concentrations, impaired GR and MR gene expression, and subsequently increased adrenal gland weights in non-stressed conditions in the offspring. In addition, this study shows that maternal HPA axis dysregulation is also associated with reduced foetal growth manifested as a lower body weight and lack of catch-up growth during development. However, the offspring still exhibited catch-up growth-related detrimental effects such as impaired fasting glucose and glucose intolerance alongside insulin resistance, resembling features seen in T2DM and maternal stress studies. Moreover, offspring from prediabetic pregnancies may be at risk of early-onset prediabetes and mental health disorders associated with HPA axis hyperactivity, hypertension and T2DM in adulthood. As a results, foetal programming might also begin in prediabetic pregnancy. Moreover, due to the challenging nature of diagnosing prediabetes, this study on prediabetes during pregnancy could potentially uncover clinical indicators that are valuable in gaining a deeper understanding of the physiological changes that take place throughout pregnancy and their impact on foetal development. Therefore, monitoring maternal periconceptional health status and conditions during pregnancy, especially for prediabetic females, may be of great importance.

## Figures and Tables

**Figure 1 ijms-25-05431-f001:**
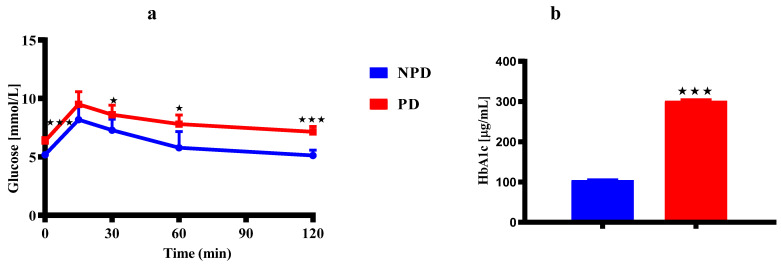
The dams’ OGTT (**a**) and HbA1c concentrations (**b**) in the NPD and PD groups (n = 6 per group) at 36 weeks. Values are expressed as mean ± SEM. ⋆ *p* < 0.011; ⋆⋆⋆ *p* < 0.001 denotes comparison with NPD. The blue bar and line graph represent the NDP group and the red bar and line group represent the PD group.

**Figure 2 ijms-25-05431-f002:**
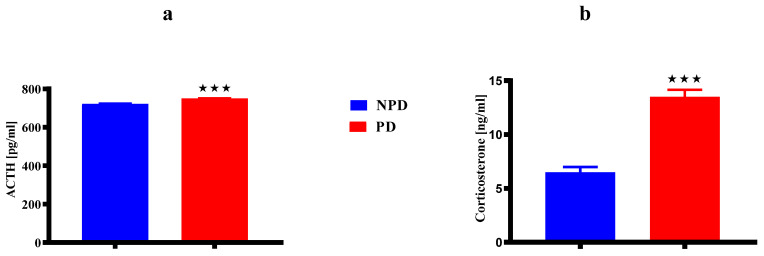
The dams’ ACTH (**a**) and corticosterone (**b**) concentrations in the NPD and PD groups (n = 6 per group) at 21 days postpartum. Values are expressed as mean ± SEM. ⋆⋆⋆ *p* < 0.001 denotes comparison with NPD. The blue bar represents the NDP group and the red bar represents the PD group.

**Figure 3 ijms-25-05431-f003:**
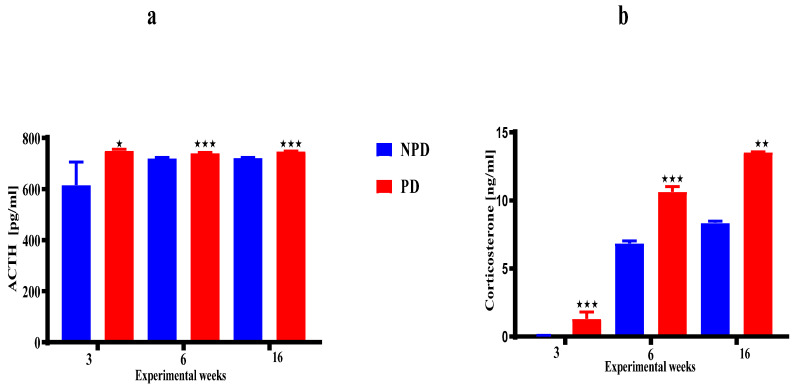
The ACTH (**a**) and corticosterone (**b**) concentrations of the pups born from the NPD group (n = 6 per week) and PD group (n = 6 per week) at weeks 3, 6, and 16 of the experimental periods. Values are expressed as mean ± SEM. ⋆ *p* < 0.0144; ⋆⋆ *p* < 0.0022; ⋆⋆⋆ *p* < 0.001 denotes comparison with NPD. The blue bar represents the NDP group and the red bar represents the PD group.

**Figure 4 ijms-25-05431-f004:**
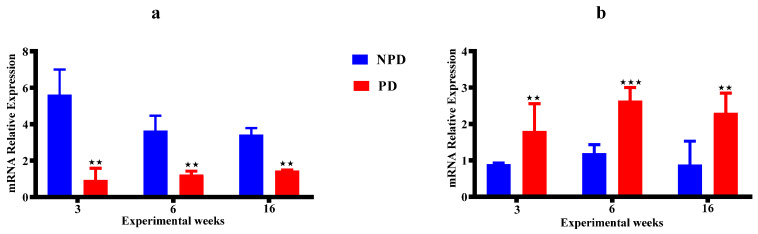
The hippocampal GR (**a**) and MR (**b**) gene expressions of the pups born from the NPD group (n = 6 per week) and PD group (n = 6 per week) at weeks 3, 6, and 16 of the experimental periods. Values are expressed as mean ± SEM. ⋆⋆ *p* < 0.0022; ⋆⋆⋆ *p* < 0.001 denotes comparison with NPD. The blue bar represents the NDP group and the red bar represents the PD group.

**Figure 5 ijms-25-05431-f005:**
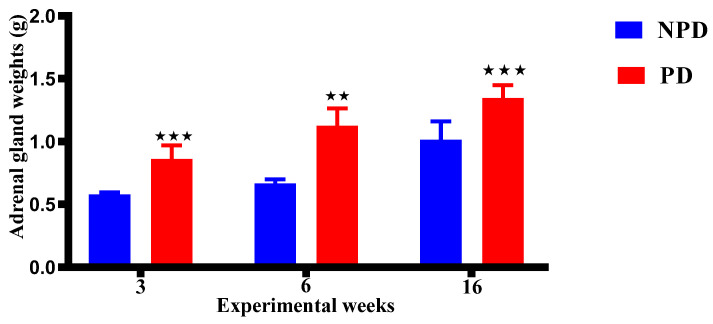
The adrenal gland weight of the pups born from the NPD group (n = 6 per week) and PD group (n = 6 per week) at weeks 3, 6, and 16 of the experimental periods. Values are expressed as mean ± SEM. ⋆⋆ *p* < 0.0022; ⋆⋆⋆ *p* < 0.001 denotes comparison with NPD. The blue bar represents the NDP group and the red bar represents the PD group.

**Figure 6 ijms-25-05431-f006:**
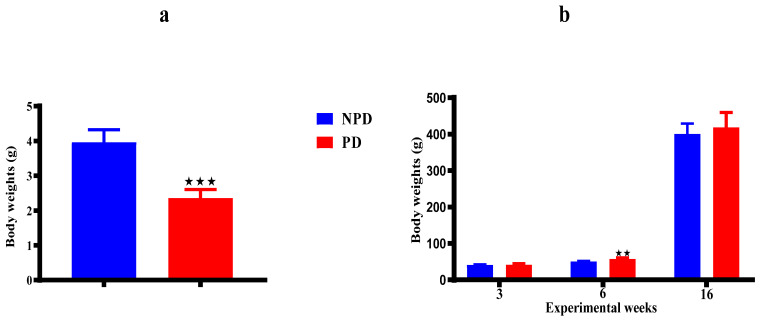
The body weights of the pups born from the NPD group and PD group on day 7 (**a**), and at weeks 3, 6, and 16 (**b**). Values are expressed as mean ± SEM. ⋆⋆ *p* < 0.0022; ⋆⋆⋆ *p* < 0.001 denotes comparison with NPD. The blue bar represents the NDP group and the red bar represents the PD group.

**Figure 7 ijms-25-05431-f007:**
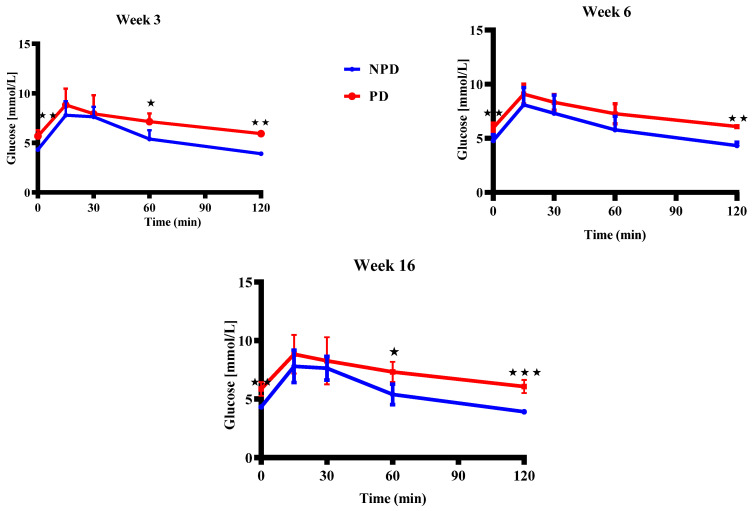
The OGTT of the pups born from the NPD group (n = 6 per week) and PD group (n = 6 per week) at weeks 3, 6, and 16 of the experimental periods. Values are expressed as mean ± SEM. ⋆ *p* < 0.0144; ⋆⋆ *p* < 0.0022; ⋆⋆⋆ *p* < 0.001 denotes comparison with NPD. The blue line graph represents the NDP group and the red line graph represents the PD group.

**Figure 8 ijms-25-05431-f008:**
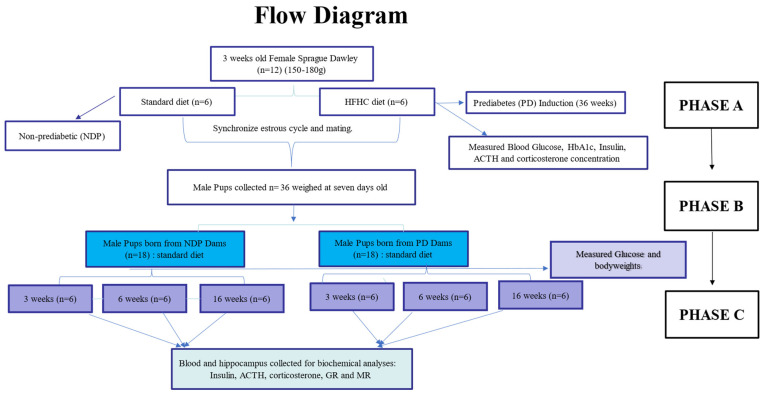
The summarised methodology flow chart in phase segments A, B and C.

**Table 1 ijms-25-05431-t001:** HOMA-IR, HOMA-S, and HOMA-β indices, and ACTH and corticosterone plasma concentrations in the non-pre-diabetic (NPD) and pre-diabetic (PD) female groups (n = 6, per group).

Groups (n = 6)	Glucose (mmol/L)	Insulin (pmol/L)	HOMA-IR	HOMA-S Values (%)	HOMA-β Values (%)	Plasma ACTH (pg/mL)	Plasma Corticosterone (ng/mL)
NPD	5.18	43.97 ± 0.08	0.84	120.80	80.10	722.3 ± 0.88	6.50 ± 0.19
PD	6.35	158.60 ± 0.45	3.06 α	32.70	126.60	750.0 ± 0.68 ⋆⋆⋆	13.51 ± 0.26 ⋆⋆⋆

Values are expressed as mean ± SEM. ⋆⋆⋆ *p* < 0.001 denotes comparison with NPD. The HOMA-IR represented as α denotes significant insulin resistance.

**Table 2 ijms-25-05431-t002:** HOMA-IR, HOMA-S, and HOMA-β indices, and ACTH and corticosterone plasma concentrations in the pups born from the non-pre-diabetic (NPD) female and pre-diabetic (PD) female group (n = 6, per group) at weeks 3, 6, and 16.

Groups (n = 6)		Glucose (mmol/L)	Insulin (pmol/L)	HOMA-IR	HOMA-S Values (%)	HOMA-β Values (%)	Plasma ACTH (pg/mL)	Plasma Corticosterone (ng/mL)
Week 3	NDP	4.3	45.89 ± 8.91	0.83	121.00	116.20	616.20 ± 40.21	0.053 ± 0.01
	PD	5.7	132.80 ± 10.80	2.51 α	39.80	137.10	751.00 ± 2.36 ⋆	1.30 ± 0.21 ⋆⋆⋆
Week 6	NDP	4.7	53.51 ± 5.13	0.98	101.60	107.70	720.80 ± 1.07	6.85 ±0.072
	PD	5.9	196.70 ±11.91	3.69 α	27.10	168.90	741.20 ± 1.07 ⋆⋆⋆	10.65 ± 0.15 ⋆⋆⋆
Week 16	NDP	4.3	94.60 ± 25.18	1.68 α	59.50	188.60	722.10 ± 0.83	8.3 ± 0.05
	PD	6.1	233.40 ± 23.36	4.37 α	22.90	179.10	748.00 ± 0.44 ⋆⋆⋆	13.5 ± 0.01 ⋆⋆⋆

Values are expressed as mean ± SEM. ⋆ *p* < 0.01; ⋆⋆⋆ *p* < 0.001 denotes comparison with NPD. The HOMA-IR represented as α denotes significant insulin resistance.

**Table 3 ijms-25-05431-t003:** Postnatal developmental stages in rats and humans [213,214,215].

Developmental Stages in Rat/Human	Rat (Male)	Human
Neonatal/newborn	0–7 days	0–28 days
Infantile/infant	8–20 days (1–2 weeks)	1–23 months
Juvenile/child	21–32 days (3–5 weeks)	2–12 years
Peripubertal/adolescent	33–55 days (5–8 weeks)	12–16 years
Late puberty/adolescent	56–70 days (8–10 weeks)	
Emerging adulthood	70–150 days (10–21 weeks)	18–25 years

Adapted from Barrow et al., 2011 [213], Picut et al., 2015 [214], and Parker and Picut, 2016) [215].

**Table 4 ijms-25-05431-t004:** List of primers used in the study.

Sequence Name	Sequence
Glucocorticoid receptor (Nr3c1 gene)	Forward: 5′-ACCTCGATGACCAAATGACC-3′ Reverse: 5′-AGCAAAGCAGAGCAGGTTTC-3′
Mineralocorticoid receptor (Nr3c2 gene)	Forward: 5′-AAAGGGTAGTGTGTGCAGGG-3′ Reverse 5′-GTTCTCCTAGTTCCCGGAGG-3′
Glyceraldehyde-3-phosphate dehydrogenase (GAPDH)	Forward: 5′-AGTGCCAGCCTCGTCTCATA-3′ Reverse 5′-GATGGTGATGGGTTTCCCGT-3′

## Data Availability

The datasets generated during the current study are available from the corresponding author on reasonable request.

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
