# Peer review of "Pregestational Prediabetes Induces Maternal Hypothalamic–Pituitary–Adrenal (HPA) Axis Dysregulation and Results in Adverse Foetal Outcomes"

_ijms, 2024, doi:10.3390/ijms25105431_

Round 1

Reviewer 1 Report

Comments and Suggestions for Authors

I have read and analyzed the manuscript from Ngema and coauthors. In my opinion, the manuscript is devoted to interesting and prospective theme. However, the manuscript raises some questions which should be resolved before publication.

1.Which prediabetes criteria authors have used? The presented animal model considered in literature as a metabolic syndrome model. Did authors perform blood pressure measurement?

2.All figure legends should be modified according to traditional rules without "...figure shows..."

3.Why authors did not perform insulin tolerance test?

4.Figure 2. ACTH graph. Is this small difference significant? If yes authors should remake the graph to graph with gap on Oy axis.

5.Can authors show melting curves for PCR products? It is my requirement as a reviewer just for the manuscript quality analysis, this data should not be added in the manuscript or supplement.

Author Response

Dear Reviewer 1

We sincerely appreciate the reviewer's invaluable time and insightful comments on our manuscript. Your thoughtful feedback has greatly enriched our work and helped us improve its quality. We are grateful for your dedication to advancing knowledge in our field.

The scientific advice provided by the reviewer was considered and appropriately handled. All authors participating in the manuscript ensured that all comments were addressed appropriately.

The manuscript's corrections made by authors from Reviewer 1 comments have yellow highlights. While corrections from reviewer 2 comments are highlighted in blue.

Reviewer 2 Report

Comments and Suggestions for Authors

Dear author’s 

I was pleased to review your interesting experimental article entitled “Pregestational prediabetes induces maternal hypothalamic-pituitary-adrenal (HPA) axis dysregulation and results in adverse 3 fetal outcomes: influences on HPA axis and development” and i have the following comments:

1. The title of the article is too long. I suggest you to review the title. For ex “

Pregestational prediabetes induces maternal hypothalamic-pituitary-adrenal (HPA) axis dysregulation and results in adverse fetal outcomes”

2. In the Introduction it is mandatory to explain the aim of the study.

3. Methods:

a) explain the type of your article.

b) why the sample of the study was so limited. In order to draw conclusion it is mandatory to enroll a relevant sample study population.

c) there is established some inclusion criteria?

d) a flowchart with the rats and groups study will be informative

3. Discussion 

a) in this section it is mandatory to compare your results with the existing literature results.

b) at the end of this section you should explain the limitations of your study.

c) can we extrapolate this results tu humans? There are studies with this endocrine disorders in pregnancy?

The main question of your study should be explain in order to make clear the study research results.

English and punctuation edits.

Author Response

24 April 2024

To the Reviewer 2

We sincerely appreciate the reviewers your invaluable time and insightful comments on our manuscript. Your thoughtful feedback has greatly enriched our work and helped us improve its quality. We are grateful for your dedication to advancing knowledge in our field.

The scientific advice provided by the reviewer was considered and appropriately handled. All authors participating in the manuscript ensured that all comments were addressed appropriately.

The manuscript's corrections made by authors from Reviewer 2 comments have blue highlights. While corrections from reviewer 1 comments are highlighted in yellow.

Round 2

Reviewer 1 Report

Comments and Suggestions for Authors

Many thanks to the authors, they have resolve all raised questions

Reviewer 2 Report

Comments and Suggestions for Authors

Dear author’s

I was pleased to review your revised article and i have the following comment:

Please place the “Study limitation and future recommendations” before conclusion.
